# Self-Boundary Dissolution in Meditation: A Phenomenological Investigation

**DOI:** 10.3390/brainsci11060819

**Published:** 2021-06-21

**Authors:** Ohad Nave, Fynn-Mathis Trautwein, Yochai Ataria, Yair Dor-Ziderman, Yoav Schweitzer, Stephen Fulder, Aviva Berkovich-Ohana

**Affiliations:** 1Department of Cognitive Sciences, Hebrew University of Jerusalem, Mt. Scopus, Jerusalem 9190501, Israel; 2Edmond Safra Brain Research Center, Faculty of Education, University of Haifa, Haifa 3498838, Israel; mathis.trautwein@uniklinik-freiburg.de (F.-M.T.); yairem@gmail.com (Y.D.-Z.); schweitzyo@gmail.com (Y.S.); 3Department of Psychosomatic Medicine and Psychotherapy, Faculty of Medicine, University of Freiburg, 79085 Freiburg im Breisgau, Germany; 4Institute for Frontier Areas of Psychology and Mental Health, 79098 Freiburg im Breisgau, Germany; 5Psychology Department, Tel-Hai Academic College, Qiryat Shemona 1220800, Israel; yochai.ataria@gmail.com; 6The Integrated Brain and Behavior Research Center (IBBRC), University of Haifa, Haifa 3498838, Israel; 7Department of Learning, Instruction and Teacher Education, Faculty of Education, University of Haifa, Haifa 3498838, Israel; 8The Israel Insight Society (Tovana), Kibbutz Ein-Dor, R.D. Izrael 1933500, Israel; stephenfulder@gmail.com; 9Department of Counseling and Human Development, Faculty of Education, University of Haifa, Haifa 3498838, Israel

**Keywords:** self boundaries, minimal self, self-dissolution, neurophenomenology, empirical phenomenology, meditation, self-transcendence

## Abstract

A fundamental aspect of the sense of self is its pre-reflective dimension specifying the self as a bounded and embodied knower and agent. Being a constant and tacit feature structuring consciousness, it eludes robust empirical exploration. Recently, deep meditative states involving global dissolution of the sense of self have been suggested as a promising path for advancing such an investigation. To that end, we conducted a comprehensive phenomenological inquiry into meditative self-boundary alteration. The induced states were systematically characterized by changes in six experiential features including the sense of location, agency, first-person perspective, attention, body sensations, and affective valence, as well as their interaction with meditative technique and overall degree of dissolution. Quantitative analyses of the relationships between these phenomenological categories highlighted a unitary dimension of boundary dissolution. Notably, passive meditative gestures of “letting go”, which reduce attentional engagement and sense of agency, emerged as driving the depth of dissolution. These findings are aligned with an enactive approach to the pre-reflective sense of self, linking its generation to sensorimotor activity and attention-demanding processes. Moreover, they set the stage for future phenomenologically informed analyses of neurophysiological data and highlight the utility of combining phenomenology and intense contemplative training for a scientific characterization of processes giving rise to the basic sense of being a bounded self.

## 1. Introduction

*Live as such: Stretch a hand into the endless**outwards of the world**and turn the outside in**and the world into a chamber**and God into a little soul**inside the endless body*–Yehuda Amichai [1].

The sense of self is familiar and robust as much as it is elusive. As a fundamental aspect structuring our conscious lives around a sense of being someone immersed in an external world, it is often argued that the sense of self has a direct bearing on how we understand and study consciousness (e.g., [2,3,4,5]). However, it persistently defies consensual definition and is notoriously challenging to make scientifically tractable. Several overlapping theoretical views emphasize a distinction between two main aspects of the sense of self, which has fruitfully guided empirical research: reflective self (‘me’) and pre-reflective self (‘I’) [3,6,7,8,9]. The reflective self (closely related to the narrative self) involves an explicit awareness of one’s perceptual image or mental state, giving rise to an enduring sense of identity (such as when contemplating one’s own thoughts, motivations, personality, memories or appearance). The second, pre-reflective self, overlapping with the minimal self, refers to oneself as a subjective knower and agent in an immediate first-personal mode of embodied presence in which experience is given. The features that are often said to characterize this fundamental sense of embodied selfhood are a sense of spatiotemporal location, ownership, agency, and an egocentric, body-centered perspective (see [5,10,11,12,13,14,15] for different conceptualizations). The pre-reflective self establishes a subtle though ubiquitous dynamic boundary, distinguishing our lived bodies, our actions and our phenomenal interiority from the external world [16].

Due to the incessant and implicit nature of the pre-reflective self, operationalizing it for empirical research is inherently difficult. Research into self-disrupting psychopathologies [17,18,19], alongside experimental manipulations [17,20,21,22,23,24] are insightful but limited to the study of local alteration and disruptions of single features of self-experience. By contrast, accumulating empirical evidence suggests that deep meditative states (as shown by our studies: [25,26,27]) involve a global dissolution of the sense of self, and the pre-reflective self in particular [28]. In addition to meditators’ claimed proficiency in experiential awareness [29,30], their specific meditative skills in generating experiential states of global self dissolution render them a uniquely apt cohort for a neurophenomenological study of the pre-reflective self. Neurophenomenology is a research paradigm set to pave a methodological path for bridging the ‘explanatory gap’ in our understanding regarding the integration of first-person phenomenological and third-person physiological features of the mind. For a recent review, see [31].

In previous studies at our lab, profound alterations of the pre-reflective self in meditative experience were explored while focusing on the sense of boundaries (SB) and its dissolution. Although intuitively considered in relation to one’s corporeal body boundaries, the SB should not be understood as a fixed and separating body-world demarcation. Conversely, as David Abram [32] (p. 38) writes:

“The boundaries of a living body are open and indeterminate; more like membranes than barriers, they define a surface of metamorphosis and exchange. Considered phenomenologically—that is, as we actually experience and live it—the body is a creative, shape-shifting entity. […] Far from restricting my access to things and to the world, the body is my very means of entering into relation with all things.” In phenomenology, the term ‘lived body’ (Leib) is used to denote the body as it is directly experienced through the inherent embodiment of oneself, as opposed to the body considered externally as a physical object (Körper) [33].

The construct SB can be understood as involving a dynamical self/non-self distinction characterizing the sense of self as a bounded being immersed, related but distinguished in its lived body from the external environment. The SB is said to be flexible in that it may open, close, shift, and extend beyond the body, structuring awareness dynamically in various domains based on actual or potential sensorimotor interactions with the environment [16,25,34]. The elemental role of boundaries in the structure of (self-)experience is evidenced in both psychopathology and meditation. Profound changes in the SB experience often accompany various psychopathologies such as complex post-traumatic stress [35] and depersonalization disorders, but may also result from prolonged meditative practice [19,28,30]. However, the current lack of knowledge about the concrete phenomenology of these experiences, which uniquely alter self-experience, hinders meticulous empirical investigation [36]. As part of a comprehensive neurophenomenological research project centered on SB dissolution, we here present the largest-to-date phenomenological study of the SB, addressing this gap in the literature of the self.

Importantly, this research is the latest in a series of studies conducted over the past decade by our research team as part of an evolving effort to render the sense of self and its dissolution scientifically tractable (described in length in [31]). The earlier studies provided phenomenological support for the notion that meditators can strongly alter their SB in meditation [26], and the neurophysiological results showed that these alterations were mediated by neural regions, notably the posterior cingulate cortex, and the temporo-parietal junction [27], linked in other studies with embodied self-processes [24,37,38,39,40,41]. The phenomenology of SB dissolution entailed a reduction in sense of agency, sense of ownership, self-location, body sensations, first-person perspective, sense of time, and self-other distinction [25]. However, these phenomenological findings were based on a single highly adept practitioner raising a concern about generalizability. In a current project, of which the presented investigation is one part, we therefore aim to extend the scope by studying a large group of long-term meditators with a specific preparatory training, while deepening and systematizing the phenomenological characterization. The overall aims of the project are replicating our proof-of-concept neurophenomenological results, specifying underlying neurocognitive models explaining the experiential categories and neural results, and exploring individual differences (neural and phenomenological). In particular, the first and the third of these aims require a fine-grained phenomenological clustering of the SB dissolution experience. Fleshing out the experiential dynamics of the SB experience is essential in creating a viable neurophenomenological dialogue in which first-person data may constrain, interpret, or guide the analysis of third-person data (brain and behaviour). The presented results thus constitute a first step towards the upcoming neurophenomenological integration by laying out the phenomenological features of SB, showing their relations and dynamics, and in this way operationalizing a phenomenologically-guided framework for subsequent neurophenomenological analyses.

Furthermore, beyond generalizing the previous results and systematizing experience characterization in a way that can guide neural analysis, we also inquired into the heterogeneity and dimensional structure of SB dissolution experiences, the dynamics of their induction, their affective varieties as well as their reliance on previous meditative practice. Thus, we also addressed the following questions: Are there distinct clusters of qualitatively different “dissolvers” or does the dissolution unfold along a single dimension of depth? What are the meditative techniques or mental micro-gestures [42] contributing to dissolution, and conversely, what kind of activity sustains the SB? Are there different varieties of affective experience associated with dissolution (as suggested by previous research [36,43]) and how does affective valence relate to SB alteration? And finally, is the depth of dissolution related to the amount of previous meditation practice? The strategy of the phenomenological investigation in approaching the foregoing questions was to exploit the inherent experiential diversity of our large cohort of meditators, rather than disregard and average it out.

To that end, we employed a novel study design with a mixed-methods approach featuring various first-person methodologies. First, we undertook an in-depth qualitative phenomenological investigation, which is based on the presupposition that any exploration of individual experiences becomes valuable through the emergence of generic structures from the analysis of multiple descriptions of experiences [34,35]. Then, we attempted to formalize the phenomenology by mapping the resulting phenomenological categories to continuous dimensions and employing quantitative analyses.

## 2. Methods

### 2.1. Participants

Forty-six proficient meditation practitioners (Aged 26–72, mean age = 39.8 ± 10.9, 19 females and 27 males, all Caucasian, 33 have academic education) with a wide variance in meditation expertise (115–24,837 h, mean = 3832 ± 4845) participated in the study between November 2018 and November 2019. Participants were mostly recruited through Tovana (The Israel Insight Society) in a convenience sample with the criterion of participation in at least one meditation retreat and a minimum of 1 year of practice. Exclusion criteria consisted of conditions that limit MEG data quality (dental splints, artificial cardiac pacemakers), active psychiatric disorders, current psychiatric medication, not having normal or corrected-to-normal vision and hearing. The research was approved by the Institutional Review Board of the Education Faculty, University of Haifa, Israel.

### 2.2. Training

Prior to the study, all practitioners participated in a 3-week specially tailored meditative training (for more details, see Appendix A: Self-boundary Meditation Training Program) developed and guided by Dr. Stephen Fulder, the founder of Tovana, a proficient meditation practitioner and teacher in the Theravada Buddhist tradition, and co-researcher in earlier neurophenomenological studies [25,27]. The training consisted of a full day workshop with guided meditation exercises and practical and theoretical discussion of the practice, as well as two afternoon sessions for additional practice and clarifications. Additionally, participants received recorded instructions for daily home practice. Presentations and meditation instructions from the workshop were videotaped for documentation and were provided to those participants that could not attend one of the meetings (see Appendix A: Self-boundary Meditation Training Program).

The training was implemented for the purpose of increasing the meditators’ prospects of successfully producing clear and stable dissolution experiences in the lab, as well as creating a dialogue from which mutual phenomenological language for the experiential exploration of the sense of boundaries could emerge. Crucially, the training focused on various meditative techniques that manipulate one’s sense of boundaries, allowing participants to personally choose and suggest accessible methods according to their own preference and previous experience. In this way, the training approach was aimed to fully utilize each participants’ own abilities, while also preparing for the unusual conditions of practicing meditation in a lab setting, involving quick alternation between meditative states and an unorthodox horizontal meditative posture suitable for magnetoencephalography (MEG) measurement.

### 2.3. Procedure

Following the training, the participants underwent a MEG session at Bar Ilan University, in which state effects were measured, comparing a non-meditation resting baseline and two meditative states of sense-of-boundaries (SB) alteration: an active control condition termed ‘with boundaries (SB+, defined as maintaining a clear sense of body boundaries)’ and the target condition termed ‘without boundaries (SB−, defined broadly as a dissolution of one’s sense of boundaries)’. The task included two consecutive blocks of volitional SB alteration. In the beginning of each block, 30 s of preparation were given to settle into the meditative state, followed by five 1-min epochs alternating between the two conditions of SB manipulation. After each block, participants rated the degree of quality and stability of the intended meditative state of each of the epochs. In order to reduce distractions to the meditation and minimize demand characteristics, we emphasized the ratings are secondary in importance and asked the meditators to refrain from actively keeping in mind the impression of their performance during the meditative session. Instead, it was stressed that any experienced meditative state, whether successful or not in producing the intended states, is of interest to the study. Note that the MEG is set up in a quiet, dark and heavy magnetically-shielded room. It is non-invasive and there is no interference from the equipment during the experiment. These factors allow creating a relaxed and intimate environment suitable for the generation of deep meditative states.

Immediately after that, the lab procedure included a battery of tasks and several questionnaires. Briefly, we started with the main SB alteration task (the focus of the present phenomenological investigation), followed by a simple agentic task exploring whether N1 suppression (assumed to be a correlate of agency) would be modulated by the two conditions of SB alteration [44,45]. The third and last task assessed neural responses to multisensory (audio-visual) stimuli, where integration effects are assumed to map the self within the peripersonal space [46]. These MEG measurements target candidate mechanisms potentially involved in SB dissolution. The scales included the study are the altered states of consciousness (5D-ASC) [47], anxiety using the State and Trait questionnaire (STAI-S) [48], and dissociation using the dissociative experience scale (DES) [49], which will be detailed elsewhere. The MEG measurements were directly followed up by a phenomenological interview concerning the experience in the SB alteration task.

### 2.4. Interview Method

The interviews were conducted based on the open-ended and iterative questioning guidelines of the micro-phenomenological interview method [50], which facilitates richly detailed descriptions of experience, and reduces subjective biases, while enabling a rigorous investigation of pre-reflective attentional, affective, and sensorial aspects of concrete moments of experience [51,52,53]. We have made use of this interview technique liberally and modified it to our experimental setup, as explained below.

The interviews lasted 21–57 min (mean = 33 min), and were conducted in a separate and quiet room. Each interview was divided into two parts (with boundaries [SB+] and without boundaries [SB−]) in which participants were assisted in sustaining their attention on specific instances of each of the experimental conditions. The interview began with a brief explanation of the method and motivation, focusing the interviewee on accurately and carefully describing what was actually experienced in the preceding meditation session and not their beliefs or theoretical ideas about such experience. Participants were directed to focus on one specific moment of receiving the audible instruction to enter the meditative state, and were then asked to unfold what was experienced immediately afterwards. Answers (‘X’) were often repeated by the interviewer and rephrased into questions that inquire in more detail (e.g., ‘How is it like to experience X’ or ‘When X happened, what else did you feel?’). Questioning in such a way minimizes inducing content and helps interviewees gain better access to their lived experience by supporting a state of “evocation”, in which past events are brought into awareness passively and are remembered more vividly [50,51].

Based on findings from earlier studies, the questioning was thematically structured in order to address several experiential themes based on the 9 categories that were previously highlighted as prominent in the experience of SB dissolution [25], and are also examined as related to candidate underlying neurocognitive mechanisms. Structuring the interviews focused and limited the scope of each interview which was deemed necessary due to the large cohort of participants in the study. However, the questioning was typically open and non-specific as to enable novel insights to come up from the meditators and refine the understanding of the experience of self-boundaries, bringing focus to other experiential themes as well as dynamic aspects of the unfolding of SB dissolution. All interviews were recorded on audio or video and transcribed verbatim with an emphasis on preserving para verbal communication [42]. All interviews were conducted in Hebrew, the native tongue of the first author and all participants except two, with which the interviews were conducted in English. Excerpts presented in this publication were all translated by the first author.

### 2.5. Analysis of Interviews

We first set out to phenomenologically characterize our participants’ experiences systematically. In doing so, we attempt to delineate common phenomenal categories characterizing the experiential landscape of typical meditative states described by study participants. This process involves a reduction of complex qualitative information, transforming it into variable categorical or quantitative data, which can be integrated with other third-person measures and guide further investigation [29,31,54]. To this end, the descriptive accounts of meditative experiences were processed through three subsequent stages of analysis which follow guidelines of qualitative analysis [55] using specific micro-phenomenological analysis tools [42,56]:1.The textual data were organized thematically, based on a preliminary list of thematic categories which were addressed in earlier studies (sense of agency, sense of ownership, self-location, body sensations and first-person perspective, sense of time, self-other distinction [25]). A sample of 20 interviews chosen randomly was coded according to these themes by the first author and two research assistants;2.The coded descriptions of each theme were examined separately across all participants in search of commonalities (concrete example follows). Each description (excerpts of one to three sentences) was abstracted and rephrased into short descriptive units capturing the essential meaning of each description. In a process of open data-driven coding, these abstracted descriptive units were clustered into a few specialized subcategories capturing the range of experiences pertaining to each experiential category. This process was done separately for both meditative conditions of SB+/SB- and for each salient category. In this stage, it became evident which categories and subcategories were robust and prevalent in the interviews. Some subcategories were conjoined, while others dropped (see Section 3);3.The participants were characterized according to the resulting categories and subcategories. Based on the full interview transcription, a numerical value was assigned to the subcategory best reflecting the type of experience described for each experiential category. This was done for each meditative condition separately (SB+\SB−). To ensure intersubjective reliability, each participant was characterized by two independent raters (out of a total of four raters including the first and last authors). Additionally, raters evaluated on a 1–10 scale the degree of SB dissolution for each participant. The characterization was done based on the recorded and transcribed interviews in full. Each assigned value was accompanied by a chosen illustrative excerpt from the interview. Prior to the characterization, raters familiarized themselves with the concepts related to the experiential categories and with guidelines of phenomenological analysis. Raters followed concrete guidelines for a critical examination of descriptions for authenticity and particular relevance to actual lived experience, as opposed to general utterances of beliefs and preconceptions [42].

Interrater agreement was assessed using the fuzzy kappa, suggested to be suitable for fuzzy datasets that include multiple choices for each category [57]. Similar to Cohen’s Kappa method, it calculates the level of agreement between raters, while taking into account agreement expected by chance, thus considered more reliable than a simple calculation of agreement ratio. Fuzzy kappa was calculated for each category and condition.

Finally, for facilitating subsequent neurophenomenological analyses, the values assigned for each participant and category were compared by the two raters and a consensual evaluation—yielding one numerical value—was achieved through discussion.

### 2.6. Quantitative Measures

Index of meditative expertise: Participants filled out a table detailing the amount of previous retreat practice (for each retreat, days in the retreat, and number of hours practiced per day) as well as home practice (for different periods of home practice, estimated dates, and minutes per day practiced). Based on this data, the sum of practiced hours was calculated for each participant as a proxy of meditative expertise.Self-ratings: After each meditation epoch, participants reported the degree of quality and stability of the respective state on a 3-point Likert scale (quality: ‘To what extent did your boundaries dissolve\appear clear?’; stability: ‘How stable was the meditation?’, 1—not much, 3—greatly). While these ratings will be used in an epoch-specific manner for the neural analyses, we computed an average for each measure in order to assess individual differences.Phenomenological dimensions: Based on the subcategories’ ordering that emerged from the qualitative analysis of the interviews, we derived a numeric measure for each subcategory, signifying an ordinal ordering within each category. Multiply assigned subcategories were averaged (see quantitative results Section 3.3 for details).

### 2.7. Quantitative Analyses

Quantitative analyses were conducted using the R statistical programming environment [58]. To assess associations between the different phenomenological dimensions, as well as between meditative expertise, self-ratings, and phenomenological dimensions, we used Spearman correlations (*r_s_*), given the ordinal nature of the measures. The qgraph package [59] was used for depicting network relationships in the data and calculating a graph analytic measure of centrality (node strength, corresponding to the weighted sum of all the significant edges of a node). Furthermore, we employed a principal component analysis to explore dimensionality of the phenomenological dimensions. A general linear model was used to test effects of different techniques on overall dissolution and the sign test for ordinal data to compare levels of affective valence between conditions.

Throughout, alpha error is controlled at *p* = 0.05 and Bonferroni corrections are applied when using multiple measures to assess the same question.

## 3. Results

“Now I can let go of the barrier and then the sensation extends... [Gesture: arms open wide and away from the body] there’s no sense of boundary, but my boundary expands and expands. It’s not *mine*, really, but... there’s dissolution happening in *something* which is a *space*. [....] It’s not like you’re non-existent but you’re part of something, a flow, an energy, light, wave. [3 s pause] But there’s no meaning to your form, to light. There’s no form.” (#6)

This exalting description of boundary dissolution is one small excerpt of the 46 conducted interviews lasting over 25 h and including over 120,000 words. The phenomenological analysis resulted in a structured characterization of each participant’s experience under each of the two meditative conditions according to eight experiential categories. In presenting the results, we will walk on two distinct but not antithetical paths: (1) a systematic mapping of the reported experiences intended for neurophenomenological integration, and (2) a descriptive phenomenological account that is qualitative, concrete, and illustrative. Both of these approaches are empirical and based on the conducted interviews, but while the first aims for generalization and quantification, the second explores more fine-grained trends in the data which are helpful in understanding the various gestures and processes that shape and constitute the sense of self in this condition. These two approaches relate to different types of bridges of a neurophenomenological dialogue [31], which here exemplify the role of a phenomenological account to constrain or guide the exploration of the accompanying neural investigation.

Below, we first present the self-reported epoch-wise results distribution; second, the phenomenological categories are detailed, and their application to each of the meditative states is mapped out; finally, quantitative analyses on the correlative and dimensional structure of the phenomenological findings are presented.

### 3.1. Quantitative Validation of Methodological Approach

#### 3.1.1. Self-Rating of Depth and Stability of Meditative States

Analysis of epoch-wise self-ratings of stability and quality of the produced states indicated that participants mostly felt successful in producing these states with a moderate to high degree of self-reported quality and stability (on a 1–3 scale mean ratings were: quality SB− = 2.31; quality SB+ = 2.58; stability SB− = 2.22; stability SB+ = 2.36; see Appendix A). There was also a considerable amount of variability in the self-ratings (see Appendix A), as well as a strong significant correlation between meditative expertise and stability of SB− (*r*_s_ = 0.60, *p*_corr_ < 0.001), but neither for SB+ ratings (*r* < 0.21, *p*_corr_ > 0.7) nor for quality of SB− (*r* = 0.24, *p*_corr_ = 0.43) (Appendix A). Informally, many participants also reported that being prompted to rate their experience did not affect the quality of meditation. These data establish that the meditators were able to enter and stably hold the respective SB states on demand and under laboratory conditions. In addition, these ratings will inform subsequent neural analyses by providing epoch-wise information in addition to the interviews’ overall classification.

#### 3.1.2. Interrater Agreement

Each participant was phenomenologically characterized in the final coding stage of analysis by two independent raters to ensure intersubjective validity. Interrater reliability marks the level of agreement between several raters involved. It was calculated for each category and condition and resulted in an average kappa of κ = 0.69, ranging from κ = 0.49 to κ = 0.79. These results demonstrate significant agreement beyond chance (see Appendix A for details). Following [60], the observed agreement can be considered moderate to substantial and thus supports reliability of the current analysis, as well as, more generally, the feasibility of reliably classifying such intricate and subtle descriptions.

### 3.2. Phenomenological Characterization

The phenomenological analysis yielded seven categories: six categories mapping the SB experience including the sense of agency, self-location, first-person perspective, attentional disposition, affective valence, and body sensations (See Table 1 or Appendix A for the glossary used for characterization); and one category describing the different techniques the meditators employed for entering the SB states (See Table 2). In addition, raters were requested to provide a holistic qualitative measure (to be compared with a quantitative measure based on the categories, see Section 3.3) evaluating the participants dissolution degree (in the SB condition only). Each category is specified by three to five subcategories indicating the possible types of experiences instantiated in that phenomenal domain. Each participant was characterized according to this categorical classification (as presented in the next sections).

It is important to emphasize that the categorization scheme is by no means exhaustive of the intricate and multivariate phenomenology described in the interviews. However, the categorization scheme captures the diversity and phenomenal richness of the different SB experiences in a way that both organizes and makes sense of the large number of descriptions collected according to several robust phenomenal domains. In this way it allows future integration with third-person neural and behavioral measures.

**Table 1 brainsci-11-00819-t001:** Experiential categories used for characterization of SB states (both SB+ and SB−).

Experiential Category	Subcategories	Experiential Category	Subcategories
Sense of Agency	1. Active (Continuously) 2. Responsive/intermediate (Task maintenance) 3. Passive (non-doing)	Attentional disposition	1. Focused and dynamic 2. Wide and dynamic 3. Wide and static 4. Formless
Self-location	1. Within body (only) 2. Body and close surrounding 3. Expansion into vast space 4. Indeterminate self-world structure 5. Other alteration (ambiguous space)	Body sensations	1. Prominent, distinct 2. Indistinct bodily sensations 3. Imperceptible, non-local
First-person perspective	1. Normal 2. Intermediate 3. Non-dual state	Affective valence	1. Highly negative 2. Slightly Negative 3. Mixed (Various opposing emotions) 4. Neutral 5. Positive

The first category is the *sense of agency*. Typically, it is defined as the sense of being in control, causing or initiating an action [9]. Although often it specifically relates to the ability to control the body, in a static meditation as explored here, it may either refer to a feeling of unfulfilled potential to move [25], a sense of effort or the ongoing ability to control attention and deliberately manipulate mental processing (sometimes termed attentional and cognitive agency; [61,62,63]). In this context, the category reflects the participants’ sense of agency ranging from being in deliberate and continuous control of attention, to complete passivity and letting go of control.

The category of *self-location* encompasses the spatial structuring of awareness in relation to one’s own location within it. In other words, it reflects the participants’ report regarding their sense of self-location (‘where I am’) under the experienced spatial frame of reference of the surrounding (‘what else is present out there’). Here the variability between experiences was notable and ranged from a very enclosed experience of space limited to the body, to wide open and expansive space, or a deconstruction of any spatial form and location. Some participants described several different states of self-location experienced in different times across their meditation epochs, therefore we used multiple allocations for this category to include the reported variations.

Although sometimes used interchangeably with self-location, the *first-person perspective* category was used here to mark the ongoing pre-reflective sense of being a subject distinct from objects perceived, as opposed to a non-dual state in which no such fragmentation between perceiver and perceived was experienced. This category is mostly relevant for the condition of SB−, with very few exceptions in the SB+ condition.

The category of *attentional disposition* is closely related to the former and it reflects the way participants were disposed towards objects of perception. It encompasses the broadness of the scope (or aperture) of attention, being wide or narrowly focused, as well as its temporal dynamics, that is, the extent to which the orientation of attention was stable or dynamic during the meditation. Although fundamental in the structure of conscious experience, attention is often disregarded in relation to the structure of the pre-reflective self. Here, it emerged as a category that was central in characterizing the process of SB dissolution.

The category of *body sensations* reflects the salience of sensations within perceptual experience. Rather than capturing any specific quality of sensation, it generally distinguishes participants according to the clarity and distinctness with which sensations were or were not experienced.

*Affective valence* marks the emotional tone experienced by the participants throughout the meditation from blissful or pleasant to stressful and intimidating.

The final category is not characteristic of self-experience but is specific to the meditation condition. *Meditation technique* captures the various gestures that were performed or the modality through which boundaries were experienced during the meditation. Body sensation scanning is one example, alongside the use of spatial imagination, relaxation, and turning attention outwards. About half of all participants combined more than one technique and so characterization included multiple allocation of subcategories.

**Table 2 brainsci-11-00819-t002:** Meditation techniques used for entering the SB states.

	With Boundaries (SB+)	Without Boundaries (SB−)
Meditation technique	A. Sensations scanning B. Feeling\visualising form of body image C. Dwelling within body boundaries	D. Sensations E. Turning attention outwards F. Imagination/memory G. Relaxation, release, passivity H. Other

Additional to the other categories, for each participant raters provided an integrative rating of the *degree of dissolution* (DD_R_), to be differentiated from a calculated summary score (DD_C_) introduced below. DD_R_ was used to holistically assess the experienced quality of self-boundaries and their dissolution as an auxiliary scale specific to SB−. The rating was based on a loosely defined aggregate of SB-related characteristics some (but not all) of which are addressed in the categories (described in Table 1). These included primarily the structure and distinctness of experienced space and of one’s localized sense of self and bodily form, the experienced relation with surroundings, thought processes, as well as the use of first-person pronouns, the attested stability of the described peak state, and the level of appraised authenticity of descriptions. Raters evaluated all participants based on general preliminary guidelines (see Appendix A: Phenomenological Glossary) outlining a scale for DD_R_ ranging from 1 (SB is defined and closed) to 3 (Sense of relation to the surrounding, albeit clearly differentiated) to 5 (Slight change in permeability, clarity or location of self boundaries) to 7 (Greater SB dissolution, interconnectedness or change in selfhood) to 9 (Radical SB dissolution, unified spaciousness, non-dual or formlessness).

#### 3.2.1. With Boundaries (SB+) Meditative Condition

In the SB+ condition, participants were asked to maintain a clear sense of their boundaries. Most categories evidenced relatively uniform distributions (Figure 1) as the participants described relatively similar ‘prototypical’ SB+ experiences (presented with examples in Table 3). The meditative technique used almost ubiquitously was sensations scanning, or as often referred to in the mindfulness literature, body scan meditation (Technique A, *n* = 38; for example see Meditator-28, or M28 in Table 3). Most of these participants were actively controlling attention (Agency#1, *n* = 35), which was narrowly focused on bodily sensations and moving across various locations on the body and its contact points with the immediate surrounding (Attention#1, *n* = 33) (e.g., M39). This purposeful shifting of attention increased sensitivity to sensations, which were perceived with greater clarity (Sensations#1, *n* = 42) and allowed construction of a clear sense of bodily boundaries (e.g., M38). Another strategy used here was to employ a visual or proprioceptive image of the body’s form to enhance and organize local somatic sensations (Technique B, *n* = 20; e.g., M6)

The spatial experience of most participants during the SB+ meditation was limited to the confines of the body. Awareness of the surrounding space of the room apart from the immediate objects touching the body was not reported. Most participants described their sense of self-location as experienced within their body (Location#1, *n* = 41). Some described feeling a stronger sense of location around the head or the chest, while others were more identified with the moving location of attention throughout the body (e.g., M20 and M29, respectively).

**Table 3 brainsci-11-00819-t003:** Examples representative of prototypical SB+ experience.

Subcategory	Quotations	Demonstration
Technique A-Sensations scanning (*n* = 38)	M28: I begin with breathing and I base my focus there, and then I notice different parts of the body—the hands, the head—there was some pressure there. I attend all contact points of the body with the surface and blanket, so… it’s like the boundaries are at the periphery of my body.	Maintaining a clear sense of boundaries
Agency #1- Active (continuously) (*n* = 34)		
Attention #1- Focused dynamic (*n* = 33)	M39: It’s bringing the attention to different parts of the body. And then trying to be very clear about where the boundaries are. [...] There’s a felt sense of the attention that is effortful and energetic.	Actively controlling attention
Sensations #1- Prominent (*n* = 42)	M38: A sense of lucidity. It’s clearer where I am. I am here. This is me. This is where it begins, and this is where it ends.	Increased sensitivity to sensations
Location #1- Within body (*n* = 41)	M20: Like some kind of spirit that I’m travelling with inside the body. [...] I feel only proximate space that is very very close.	Experiencing within their body
	M29: Attention is simply listening to the body, to its sensation, and that naturally forms a certain space that kind of blocks it in some border.	
Technique B- Feeling\visualizing form of body image (*n* = 20)	M6: I can use a broad view, to see myself for a second, as if I opened my eyes and looked at myself. [...] It demands some effort to stay with sensations and with this gaze… discerning, focusing at… a physical sensation that defines a shape. The bladder, there’s a shape there. It’s pressing.	Combining body scan with maintaining a visual or proprioceptive image of the body’s form

While most participants described variations of the ‘prototypical’ SB+ experience described above, several other patterns were also reported (see Table 4). A smaller group of participants did not continuously and actively control attention, but instead allowed a more spontaneous attentional activity to occur. A moderate sense of agency was still present in maintaining awareness of the appearance of mind wandering or an occasional redirection of attention to body areas in which sensations appeared less distinctly (Agency#2, *n* = 13; e.g., M41). Some of these participants also reported having a wider scope of attention that was either more dynamic (Attention#2, *n* = 9; e.g., M10) or more static (Attention#3, *n* = 4), a sense of the surrounding space that extended beyond the limits of the body (Location#2,#4, *n* = 5) and a technique that emphasized a sense of passive spacious presence within the entire space of the body (Technique C, *n* = 15) (e.g., M12).

Finally, the affective valence category exhibited the largest variability of all categories and contrasted with the SB− condition in various ways. Some experienced the meditation positively as effortless or familiar (Affective#5, *n* = 13). Interestingly, these participants often described being within boundaries as a pleasant feeling of familiarity and control (e.g., M41). Others reported their experience more negatively (Affective#1-2, *n* = 14). Meditating on boundaries was in these cases described as constraining or even slightly claustrophobic and was often accompanied with a mild feeling of tension and anxiety (e.g., M40). For a few others, simply returning to their habitual boundaries from a more blissful state of boundarylessness was experienced as somewhat disappointing. Finally, some participants reported a mix of contrasting emotions (Affective#3, *n* = 6) or a relatively neutral affective state (Affective#4, *n* = 13).

**Table 4 brainsci-11-00819-t004:** Examples of less prevalent SB+ experiences.

Subcategory	Quotations	Demonstration
Agency #2- Responsive/intermediate (*n* = 13)	M41: Sometimes [attention] was moving and it didn’t happen deliberately. Moments where I find myself in the sense of body boundaries at the scalp. It came and went on its own. When I identified consciousness wandering off from body sensations, I brought it back.	Maintaining awareness of the appearance of mind wandering and sensations
Attention #2- Wide and dynamic (*n* = 9)	M10: Feelings of weight and warmth… I passed from one area to the other, wherever there was tension, and then with attention it loosened a bit, and the feeling spread out.	Attention to areas and non-specific feelings
Attention #3- Wide and static (*n* = 4)	M12: I simply feel all my body and then it really loosens up like with the volume and the length and width. Like… like being touched all over my body [... becomes] a sense of my presence as a body in space.	Passive presence within boundaries accompanied by a wider scope of attention, inclusive of surrounding space
Technique C- Dwelling within body boundaries (*n* = 15)		
Affective valence #5-Positive (*n* = 13)	M41: That clarity [of bodily sensations] was nice. It was nice to focus on it because it was very concrete. It served as an anchor. Very… tangible.	A pleasant feeling of familiarity and control
Affective valence #2-Slightly Negative (*n* = 13)	M40: When you feel the boundaries, you are also more present to the body’s discomfort.	Bodily discomfort

**Figure 1 brainsci-11-00819-f001:**
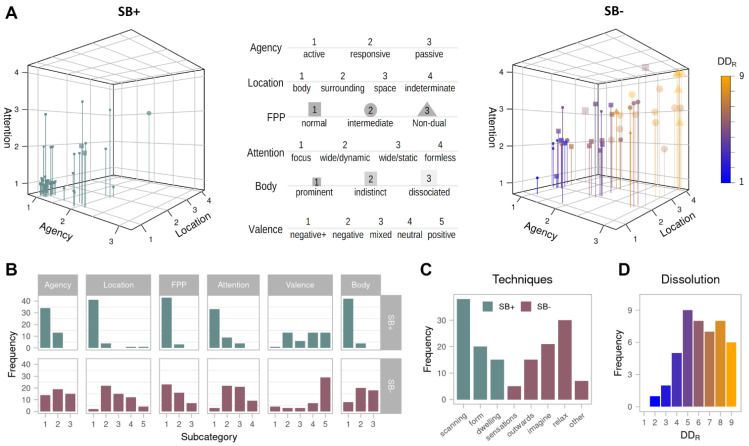
Descriptive figures of phenomenological results. Panel (**A**): Representation of the phenomenological space spanned by the Agency, Location, and Attention dimensions in the SB+ (left) and SB− (right) condition. Each data point represents one participant. Additionally, first-person perspective (FPP) is indicated by the symbol, the body sensations category by the saturation and size of markers and the rated degree of dissolution by the colour scale. Random noise was added to the individual values to make overlapping points from different participants visible. For reference, the panel shows a short version of the labels of the subcategories, for a more detailed description of the subcategories please refer to Table 1. Panel (**B**): Frequencies of occurrence of the individual subcategories. Panel (**C**): Frequency of employed techniques. Panel (**D**): Distribution of the rated degree of dissolution (DD_R_). Note that multiple assignments of subcategories were counted in panel (**D**), whereas a single value was obtained for the dimensional presentation in panels (**A**,**C**) (see methods Section 2.7 and results Section 3.3 for more details).

#### 3.2.2. Without Boundaries (SB−) Meditative Condition

The SB− meditative condition asked participants to dissolve their sense of boundaries. One striking initial result is that dissolving boundaries evidenced significantly more diversity, idiosynchronicity, and phenomenological richness than maintaining boundaries. Accordingly, the SB− section is longer and is presented with more detail and structure. Despite being often embellished with vague and figurative language, it was nevertheless possible to extract several meaningful, robust, and generic phenomenological characteristics (results are shown in Table 5, Table 6, Table 7, Table 8, Table 9, Table 10 and Table 11 and Figure 1). A sample of the numerous verbal accounts portraying the experience of boundary dissolution is presented below, followed by the categories’ systematic presentation.

“A sense of floating in some sky consciousness” (M13)\“There’s something very delicate, very innocent there, vulnerable. I can’t explain” (M32)\“An experience without an observer interfering, discerning.” (M46)\“Like leaning back inside the head” (M32)\“It’s a puddle and there’s no… it’s not solid anymore. (M11)\“some sort of intergalactic blackness [...] completely still and deeply serene.” (M43)

##### Meditation Technique

Four different meditation techniques (Table 5, Figure 1) employing various subtle mental gestures were used as tools giving rise to boundary dissolution experiences. Most participants used more than one technique. Interestingly, the gestures employed were generally viewed as aids for creating conditions allowing boundary dissolution to occur rather than actively dissolving them (e.g., M23). Accordingly, the most frequent practice method reported was that of release, relaxation, or passivity (Technique G, *n* = 29). Many of these participants reported physical loosening of muscle tension in the eyes, face, or entire body, often accompanied with deeper breathing. These gestures were not only reported but often clearly enacted during the interview (e.g., M44). This method of practice often involved a gesture of letting go of intentional effort and of the need to do something (e.g., M28). As specified in the following categories of agency and attention, this technique was exercised to different degrees with many combining this gesture of relaxation with more active techniques, while others completely disengaged from mental activity and perceptual content and surrendered all sense of ‘doing’ (e.g., M25).

The second commonly used technique was the use of imagination (Technique F, *n* = 21). Some participants imagined situations or significant past events, which helped trigger a specific mood or feeling such as love or tranquility, which promoted SB dissolution in some way (e.g., M11). Many others used spatial imagination, which often featured vast expansive landscapes, most notably the sky and outer space (e.g., M13).

Some participants based a process of SB dissolution on turning their attention outwards (Technique E, *n* = 15). Often contrasted with SB+ condition, participants became receptive to perception of what surrounds them, including room sounds (e.g., M8). Some used that to initiate a feeling of expansion outwards (e.g., M10), while others used diverting attention as a way to disengage from any perceived object (e.g., M21).

A few participants focused on their sensations (Technique#D, *n* = 5), and noticed the indistinctness of a local sense of boundary (e.g., M39).

**Table 5 brainsci-11-00819-t005:** SB− representative examples of meditation technique.

Subcategory	Quotations	Demonstration
Technique G—Release, relaxation or passivity (*n* = 29)	M23: I simply imagine my body dissolving into the mattress [...] It *is* something I’m directing but then it’s like it gets out of my control… It really happens in itself.	The meditative gestures employed create conditions which allow SB dissolution
	M44: Immediately there’s some relief… [speech slowing down] like now it’s okay to go back and expand [smiles and opens her arms]. It’s this movement of... [audible whisper of breathing out, closes her eyes and widens her arms to the sides] … Like there’s an air-conditioner working, and then it’s turned off. And then there’s some kind of... [lets her head fall back and sighs...] It’s okay to stop doing something.	Enacted gestures of relaxation
	M28: There’s effort in letting go. It’s… it’s funny to say that. There’s a sense of giving up on the holding on. But then there’s no need for more effort—you just give up.	Letting go of intentional effort
	M25: It’s like I’m falling down there. [...] Some kind of wide opening with no direction. I think there’s no other movement. No mental movement of wanting or not wanting. Some kind of presence of simply being.	Disengage from mental activity and perceptual content
Technique F—Imagination (*n* = 21)	M11: I imagined myself on one vacation [...] lying on a rhapsody looking at the water and then I just stayed with... this sense of melting of the water and this soft movement.	Imagined situations
	M13: Some kind of sky consciousness, a sort of flight… not looking downwards but sort of drifting. I was sort of floating in the sky… I saw some clouds. Something very loose, light.	Spatial imagination
Technique E—Turning attention outwards (*n* = 15)	M8: There’s more attention on the sound of the room… of breathing. There’s an experience of the body with no specific attention or specific sensations. […] It’s a sort of roaming with whatever comes up.	Open attention to drifting freely with the surrounding
	M10: I was feeling the room and then I had this feeling of awareness being in the rooms outside where you [experimenters] were, the lobby. So I felt aware of all that and the people there, and it started to open and open, I mean, to the grass outside the building and the campus.	A feeling of expansion outwards
	M21: I turn the inward gaze outwards. [...] I’m trying not to focus on anything and from that place can come something obscure and hazy which is just... nothing. Like being focused on nothing. My effort goes to keeping it there.	Disengage from any perceived object through diverting attention
Technique D—Focus on sensations (*n* = 5)	M39: So when my hand is kind of immersed in the other hand [...] There is a sense of, when you sink, as you sink into that thing, you actually kind of penetrate the boundaries of it in a certain way.	Notice the indistinctness of a local sense of boundary

##### Self-Location

Perhaps the most discriminative difference between participants regarded their spatial experience descriptions (Table 6, Figure 1). The sense of self-location changed considerably during this meditation and became much fuzzier for many participants, whereas the experience of space became more prominent. This can also be characterized as an obscuring of the distinction between self-location and the surrounding spatial frame of reference, or in other words, an experienced dissolution of the boundary between oneself and the world. The centrality and form of the body within these spatial experiences also varied significantly.

A distinguished group of participants described experiences of deep boundary dissolution, which featured an amorphous sense of space and a faint sense of self-location or lack thereof (Location#4, *n* = 12; e.g., M28 and M6). In this type of experience, participants described a radical loss of sense of self and of one’s own body. What remained was often a feeling of total immersion within space, which itself mostly lacked form, directionality or other evident features apart from references of transmodal feelings of subtle movement (such as flowing, floating, vibrating; e.g., M11 and M25). These particularly interesting experiences of deep boundary dissolution will be further explored in light of other categories and finally further analyzed in their own right, as a distinct phenomenological pattern of self-transcendence.

In stark contrast, many participants described their sense of self-location with direct reference to their body and in a strong relation to the surrounding space (Location#2, *n* = 21). Contrary to the previous subcategory, these experiences were often clearly centered around one’s body location, although during the meditation its form became less distinct. The sense of boundaries was mostly still apparent but described as less defined (e.g., M34 and M36).

Other participants described a sense of expansion into vast space (Location#3, *n* = 14). This category encompasses not only a wider spatial frame of reference but often also a sense of dynamic expansion of oneself towards it, to include it within one’s perceptual experience or one’s expanded sense of self-location (e.g., M31). While sometimes described as an expansion beginning in one’s body opening to surrounding space, this subcategory is characterized by fewer and less distinct references to one’s body and its form and a greater sense of identification with one’s experiential scope, which increasingly becomes spacious.

The most common pattern reported in the contrasting SB+ condition (being located within one’s body with minimal reference to the surrounding) was in this condition much less frequent (Location#1, *n* = 2). Finally, few participants reported other ambiguous spatial experiences which included changes in proprioception like felt deformation of body form and unstable sense of body orientation, as well as quasi out-of-body experience (Location#5, *n* = 6).

**Table 6 brainsci-11-00819-t006:** SB− representative examples of self-location.

Subcategory	Quotations	Demonstration
Location #4-Indeterminate self-world structure-radical change (*n* = 12)	M28: There’s a giving up on the body. There’s no relation to the body. Consciousness is more in peripheral space. [...] Like you’re floating. Consciousness is floating somewhere vague.	An amorphous sense of space and a faint sense of self-location or lack thereof
	M6: There’s a field of sensuality, and it’s not bounded. […] I’m not located anywhere. I am not. [...] It didn’t lose the sense of being-part-of, not entirely.	
	M11: It’s a puddle and there’s no… it’s not solid anymore. [...] There’s no longer me. It… [7 s of silence] it stops being separate. It’s not me melting on the bed. There’s no me and there’s no bed. It all melts together.	Total immersion within space
	M25: It’s like falling down there [...] Opening, release. It’s a space that’s… empty. There’s nothing in it.	
Location #2-Body & wider surrounding (*n* = 21)	M34: My attention shifted to being more in a space that… is also the body but also around the body. It’s not the universe, just around the body […] like some pleasant cloud, and its edges… not clear where it ends.	Located centered in the body, SB apparent but less defined
	M36: The sense of body boundaries is beginning to diminish and there’s a sense of intimacy or belonging that’s experienced somehow in relation to the room’s boundaries.	
Location #3-Expansion into vast space (*n* = 14)	M31: It’s as if I’m present in all of space altogether, like all of the space that my consciousness surrounds in this moment. […] It’s like my presence is something much bigger than, say, just where my body is located. Rather, who I am is present in a very large space, large at least like, say, a building, or something like that.	Dynamic expansion of oneself

##### Attentional Disposition

The widening of attention was one of the most distinctive characteristics of the SB- state (Table 7, Figure 1). In stark contrast to the focused scan frequently reported in SB+ condition, here participants described almost unanimously a widening of their attentional scope. As presented next, the description of attentional disposition did vary in its dynamics and mode of engagement (which directly relates to the sense of agency).

A large group of participants described attention as wide and dynamically active in exploration (Attention#2, *n* = 22). Some were deliberately controlling it towards certain objects of perception or of imagination, while others simply let it drift around freely (e.g., M8). All of them, however, seemed to be engaged in some process of noticing or discerning distinct features within their experience (e.g., M20).

Another group of participants similarly described an opening of the scope of attention, but distinctively reported attention as more static and stable (Attention#3, *n* = 21). These participants did not seem to be as actively engaged as the previous group but were rather receptive to whatever arises in experience (coinciding with subcategory Agency#2) (e.g., M34). Correspondingly, features in experience were reported as less defined (e.g., M39), which presumably contributed to the sense of boundary dissolution.

A radical change in attentional disposition was reported by a small group of participants who refrained from describing attention in any defined form (Attention#4, *n* = 9). These participants were exceptionally passive in letting go any sense of control over their attention (coinciding with subcategory Agency#3). Their experiences were often described as vague or formless (e.g., M33) and sometimes lacked any clear content—a state which can be referred to as ‘pure consciousness’ or ‘cessation’. This was described as a highly delicate state of total passivity and indifference that could be externally obstructed with stimuli appearance drawing attention (e.g., M32).

A few participants described a sense of focused attention (Attention#1, *n* = 3). Only one of them, an expert practitioner, reported a deep state of boundary dissolution, which is possibly the result of an extremely stable single-pointed concentration on his breath.

**Table 7 brainsci-11-00819-t007:** SB− representative examples of attentional disposition.

Subcategory	Quotations	Demonstration
Attention #2-Wide, Dynamic (*n* = 22)	M8: There’s more attention on the sound of the room… of breathing. There’s an experience of the body with no specific attention or specific sensations. […] It’s a sort of roaming with whatever comes up.	Let attention drift around freely
	M20: I’m taking my attention, this energetic feeling, to wander [...] I can feel the whole room, and sometimes I left for other spaces, sometimes to the sky, and I was kind of travelling.	Discerning distinct features within the experience
Attention #3-Wide, static (*n* = 21)	M34: Attention is much more open and wide. It doesn’t include this sense of effort of being with something. All of a sudden attention rests and things appear in it… but it doesn’t stick. There’s much less preference for anything.	Receptive to whatever arises in experience
	M39: [Attention is] wider than the body, so it kind of permeates the space around the body. [8 silent seconds] It is quite soft, it feels quite light.	Wide attention
Attention #4-Formless (*n* = 9)	M33: Attention flows with everything […] It’s nowhere but it’s also not lost. […] A kind of very very strong sensory experience of flowing and unraveling of all that was condensed.	Vague or formless experience
	M32: Attention isn’t there. It’s gone. There’s no experience at all, so I don’t know what is there. I know that at some point attention gets back to some experience but there’s a stage in which there’s no… no space, nothing.	Lacking content and form completely

##### Sense of Agency

As explained before, the sense of agency in the context of an immobile meditation involves attentional and cognitive control. It ranged from the felt sense of deliberate and continuous control of attention, to a sense of complete passivity and letting go of control (Table 8, Figure 1). This clearly relates to the previous subcategory of attentional disposition, which captured the dynamics of attention, and indeed these two categories are highly correlated (see the section on quantitative relationships between categories below). However, sense of agency in this given context relates more directly to intention and the sense of exerting volitional effort.

One group of participants described their meditation as a dynamically active process in which they were engaged in manipulating their experience in order to dissolve boundaries. This clear intention along with a continuous and to some degree effortful control of attention marks a strong sense of active agency (Agency#1, *n* = 14) (e.g., M18 and M31).

A second group of participants described conscious states that are distinctively less active. These participants did not continuously exert effort in altering their experience but only occasionally. Some tried to let go into a passive experience of dissolution but were unable to do so easily (e.g., M41). Others simply maintained an intention to notice distractions and find balance in an unstable experience of spaciousness. This way or the other, these participants still maintained a sense of agency although less distinguished and more passively receptive (Attention#2, *n* = 19).

The third group of participants described passive states in which there was little to no sense of agency reported (Agency#3, *n* = 15). These participants often described their meditation through the lack of action and control (e.g., M34), replaced by what can be described as a sense of release and surrender to the flow of experience (e.g., M6). This was also evident in the participants’ descriptions in the decreased verbal use of the first-person pronoun and of active verbs.

**Table 8 brainsci-11-00819-t008:** SB− representative examples of sense of agency.

Subcategory	Quotations	Demonstration
Agency #1-Active (*n* = 14)	M18: It’s not happening by itself, so I try to put myself in that state of expanding but then concentration is lost, so I need to activate it again.	Clear intention with a continuous and effortful sense of control
	M31: My action is to try and keep blowing up this balloon, through imagination and moving the mental gaze outwards to a wider space. [...] It’s like a struggle between this centering gravitational force, and the attempt to really push the dividing line outwards as much as I can, until it’s gone.	
Agency #2-Responsive (*n* = 19)	M41: I’m still directed towards a certain task [e.g dissolving boundaries]. It’s not this total surrender that I’m a vessel to anything that’s there. There’s an agenda… like an easy effort without ambition, yes, some sort of intention.	Somewhat passive while maintaining a task
Agency #3-Passive (*n* = 15)	M6: A deep breath… and then everything loses its grip. [...] Until now I was putting an artificial barrier over sensation, and now it’s possible to let go of this barrier and let it extend… [opens her hands away from the body]	A sense of release and surrender
	M34: There’s an element of complete inaction. I sense within it the movement of breath, but its base is very still. I feel I can characterize it as deep serenity.	Lack of action

##### First-Person Perspective

The meditative states presented here explore different senses of one’s boundaries and their dissolution, but participants often focused on their sense of bodily boundaries. A more subtle boundary reported is that which divides one’s sense of self as a perceiving subject apart from the content perceived as external. We use the category of first-person perspective to denote changes in this fundamental dual intentional structure of consciousness in which one experiences oneself as an observer of mental phenomena (Table 9, Figure 1).

While many participants did not experience changes in this aspect of their experience (1PP#1, *n* = 23), some reported what can be described as non-dual awareness (1PP#3, *n* = 7), which was sometimes referred to as a field of happening, which is neither external nor internal, and which is not relative to a first-personal observer position (e.g., M32). In these experiences, the structure of awareness pertaining to attentional and perceptual objects shifted towards a more unitive sense of open space (e.g., M6).

Other participants reported an intermediate position in which the sense of first-person perspective occasionally lost its stability, and experience ceased to be egocentrically structured (1PP#2, *n* = 16) (e.g., M26). Some also referred to it as an experience of intimacy in which there was narrowing of the felt distance or alterity between subject and object (e.g., M35).

**Table 9 brainsci-11-00819-t009:** SB- representative examples of first-person perspective.

Subcategory	Quotations	Demonstration
1PP #3-Non-dual (*n* = 7)	M32: The process begins from the inside and then there’s nothing inside or outside. [...] There’s something very delicate, very innocent there, fragile… [...] There’s no experience of attention at all. [...] No someone that’s… no attending.	Not relative to a first-personal observer position. Formless experience.
	M6: There’s a field of sensuality, and it’s not bounded. […] I’m not located anywhere. I am not. It contributes to the dissolution. There’s no sense of observer. No witness. [...] It didn’t lose the sense of being-part-of, not entirely. [...] There’s a sense of flow and it is going through something.	
1PP #2-Intermediate (*n* = 12)	M26: In the beginning there was clearly a center referring to the boundarylessness, so there was some relation between two things, but in other moments space was the dominant thing without feeling the ‘you’ relating to space.	Blurry or unstable distinction between subject and object
	M35: [In some area] there was more intimacy… [5 s pause] there is greater cohesion between the sensation and the observer.	

##### Body Sensations

The meditation in the SB− condition elicited for most participants experiences in which bodily sensations became less clearly distinct and localized (Table 10, Figure 1). As the target of attentional focus diffused from the body elsewhere, many participants described the body as peripheral in experience and its sensations indistinct (Sensations#2, *n* = 20). These participants often related to their body in some way but described sensations as airy and soft, or more generally subtle, rather than dense and pronounced (e.g., M37).

Participants who reported deeper states of dissolution of boundaries were dissociated to a greater degree from their bodily sensations (Sensations#3, *n* = 18). Some of them refrained from reporting anything specific regarding their body. Some felt the body completely dissolved or disappeared from experience. Others described general non-local feelings of floating, flowing, or vibrating, which can be termed transmodal feelings (e.g., M43) [64].

A smaller group of participants experienced their body more concretely as central in their experience, accompanied with specific and distinct sensations (Sensations#1, *n* = 8) (e.g., M39).

**Table 10 brainsci-11-00819-t010:** SB− representative examples of body sensations.

Subcategory	Quotations	Demonstration
Sensations #1-Prominent (*n* = 8)	M39: There’s an opening in the chest, opening in the shoulders, sometime a kind of opening of the face, and I’m like, two sides of the face can’t just sort of fall [extends the word] to either side, allow gravity in each side to pull its own direction	Concrete reference to local and specific bodily sensations
Sensations #2-Indistinct (*n* = 20)	M37: I was less aware of bodily experience. Just something more loose, that the body feels more like one piece.	Sensations clarity diminish
Sensations #3-Dissociated, imperceptible (*n* = 18)	M43: It’s this feeling of floating, of something sort of airy, wide open, spacious floating	Non-local feeling with no references to the body

##### Affective Valence

Most participants experienced the SB− state as a pleasant experience ranging from comfortable and peaceful to joyous and blissful (Affective#5, *n* = 29) (Table 11, Figure 1). These positive feelings were often related to a sense of soft or spacious being involving open relaxation (e.g., M13 and M42). Others chose to describe their experience with warm feelings of love and connection.

Few participants described their affective state as negative. Some of them were mildly stressed and felt a sense of unpleasant struggle and difficulty (Affective#2, *n* = 3) (e.g., M27 and M40), while others were more strongly affected with stressful fear related to an experienced loss of control (Affective#1, *n* = 4) (e.g., M7). Others reported a mix of contrasting emotions (Affective#3, *n* = 3) or a relatively neutral or equanimous affective state (Affective#4, *n* = 7).

**Table 11 brainsci-11-00819-t011:** SB− representative examples of affective valence.

Subcategory	Quotations	Demonstration
Affective #5-positive (*n* = 29)	M13: It was released, a flight. It was much much more pleasant. I realized the body is a burden and I became connected with a kind of vast consciousness.	A sense of soft or spacious being involving open relaxation
	M42: I felt more security or calm, like it’s my home. [...] Being without boundaries generated some sort of serenity, of spaciousness.	
Affective #2-Slightly negative (*n* = 3)	M27: I experienced stress in having to quickly change from a state of being entirely in the body to a state in which I’m really not in the body, supposedly. So I felt stress because I felt the time dimension. That I won’t make it.	Stress and other mild emotions
	M40: It was more emotional around this sense of being in space. A trace of anxiety… a strange thought of… preparing for what may come...	
Affective #1-Highly negative (*n* = 4)	M7: I felt my heart pounding as if I’m in a dramatic moment of my life. Really even slightly intimidating in intensity. […] I feel like being in a closed room all day, and then opening the door and realizing I’m above a jungle.	Stressful fear

##### Rated Degree of Dissolution

As explained earlier, rated *degree of dissolution* (DD_R_) was used as a higher-order evaluative category holistically assessing the experienced quality of self-boundaries and their dissolution based on a loosely defined aggregate of various related characteristics, most of which are related to the other phenomenological categories (see Appendix A: Phenomenological Glossary). As shown in Figure 1D, DD_R_ was well-distributed (mean = 6.2, SD = 1.82) indicating that participants generated experiences of SB dissolution to different degrees of depth.

### 3.3. Quantitative Relationships between Categories

The description of the SB experiential states subcategories resulting from the qualitative analysis suggested a continuum-like interpretation (e.g., sense of agency decreases from 1 = active → 2 = responsive → 3 = passive). This interpretation was further supported by the fact that multiple allocations mainly occurred for neighbouring (and residual) subcategories (e.g., some reports were classified as both responsive and passive) (of 20 multiple allocations in total, only 2 were non-neighbouring). Accordingly, the subcategories were treated as numeric data ordered on an ordinal scale for each category (see Table 1). Taken together, these scales define a phenomenal multidimensional space into which each participant could be mapped. An average value was obtained when a participant’s descriptions indicated multiple subcategories, and the “other” subcategory for self-location was dropped (it only occurred in combination with other subcategories). This transformation allowed us to plot the data in a multidimensional space (see Figure 1A) and to conduct further statistical analyses elucidating relationships between the categories. For brevity and to differentiate the dimensional interpretation from the qualitative categories, the derived dimensions are referred to in capitalized form as Agency, Location, FPP, Attention, Valence and Body. Note that the following analyses are reported for the SB− condition given the focus of the paper on the phenomenology of SB dissolution. In addition, the variance in the SB+ condition was very low (cf. Figure 1A), which would undermine the robustness of such quantitative analyses. Nevertheless, surprisingly similar patterns of association emerged for the SB+ condition (cf. Appendix A).

First, we computed mutual (Spearman) correlations between the dimensions, which yielded mostly strong and significant positive correlations, ranging from *r*_s_ = 0.50 (Agency with Location) to *r*_s_ = 0.80 (Location and Body); only Valence was not significantly correlated with the other dimensions (see Appendix A for values and plots for all correlations). Significant associations between categories are summarized in the network graph of Figure 2A, highlighting the strong intercorrelation of the main categories and the relative independence of the Valence dimension. Furthermore, we computed a partial correlation network which shows the remaining associations between nodes after controlling for all other variables in the network (Figure 2B), thus highlighting uniquely shared variances between variables and indicating potential causal relationships [65]. This analysis indicated that a collapse in FPP was related to two different “routes of dissolution”: One that combined the Location and Body dimensions and another one hinging on Agency and Attention. In this scenario (i.e., when accounting for the other dimensions), the relationship between the two central “hubs”—Location and Agency—was even antagonistic. We also derived a metric of strength centrality (depicted by the orange coloring of the nodes), corresponding to the weighted sum of all the significant edges of a node, highlighting Agency, and to a lesser extent Location, as central dimensions driving the process of dissolution on the other dimensions.

**Figure 2 brainsci-11-00819-f002:**
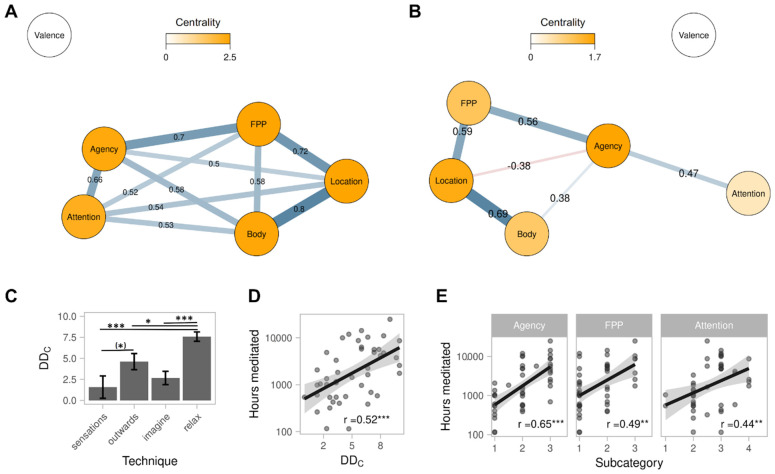
Quantitative relationships between phenomenological categories in the SB− condition. Panel (**A**): Network graph of the bivariate (Spearman) correlations between the phenomenological categories. Distances between nodes as well as the width and colouring of the edges reflect strength of association. Blue edges refer to positive and red edges to negative relationships. Edges are drawn only for significant correlations and size of the coefficients is plotted on top of the edges. Panel (**B**): Partial correlation network showing associations between nodes after controlling for all the other nodes in the network. Panel (**C**): Parameter estimates from a linear model predicting the computed degree of dissolution (DD_C_) based on the employed meditative techniques. Error bars correspond to standard errors. Sensations = scanning sensations; outwards = turning attention outwards; imagine = imagination/memory; relax = relaxation, release, passivity (* *p* ≤ 0.05, ** *p* ≤ 0.01, *** *p* ≤ 0.001). Panel (**D**): Scatterplot and correlation coefficient (Spearman) showing the association between computed degree of dissolution (DD_c_) and lifetime hours of meditative practice (on a logarithmic axis). Panel (**E**): Scatterplots for those phenomenological dimensions that correlated significantly with lifetime hours of meditative practice.

Moreover, we computed Spearman correlations between DD_R_ and the six dimensions. The results indicated that all of the dimensions, except Valence, contributed strongly and significantly to overall dissolution (all *r*_s_ > 0.66; see Appendix A). These analyses suggested that a large part of the variance in the phenomenological space that characterizes SB dissolution can be subsumed under a single dimension. In fact, a principal component analysis confirmed that 58% of the variance in all dimensions could be explained by one component, with very high loadings on all dimensions, except Valence (standardized loadings for Agency: 0.83; Location: 0.85; FPP: 0.84; Attention: 0.78; Body: 0.80; Valence: 0.33). Therefore, for subsequent quantitative analyses, we derived a summary score as the unweighted sum of all dimensions (except Valence), henceforth referred to as ‘calculated degree of dissolution’ (DD_C_). Interestingly, this score correlated very highly with the overall dissolution rating DD_R_ (*r*_s_ = 0.91). Since the latter is a holistic rating of various facets and cues indicating increasing depth of dissolution (see Appendix A: Phenomenological Glossary), the high correlation indicates that dissolution is effectively and sufficiently described by these five dimensions. DD_C_ thus provides a formalized index capturing the degree of dissolution. Followingly, this score was used to test the effect of different techniques (or mental gestures) on overall SB dissolution. To this end, we computed a linear model predicting DD_C_ based on the presence or absence of each of the techniques (corresponding to four binary predictors; the “other” subcategory was omitted due to its fuzziness). Parameter estimates for each technique as well as significance of simple contrasts are shown in Figure 2C. The technique “relaxation, release, passivity”, and to a lesser degree “turning attention outwards” clearly emerged as the strongest inducers of boundary dissolution.

Finally, given that meditation related changes in SB have been associated with a wide range of different affective responses (e.g., [36]), we tested whether affective experience in the dissolution condition (SB−) differed from the condition with boundaries (SB+) using the sign test for ordinal data. Interestingly, Affect was clearly more positive in the SB- condition compared to SB+ (*p* < 0.001; cf. Figure 1D for distributions of Affect in both conditions).

The amount of previous meditative practice is often regarded as a proxy for meditative expertise; and within the research paradigm of neurophenomenology, such expertise has been suggested to support navigating and reporting on conscious experience [53,66]. To evaluate these claims, we computed Spearman correlations between lifetime meditation practice (total hours meditated) and the different phenomenological dimensions. Results yield strong effects for the DD_c_ as well as the Agency, FPP, and Attention dimensions (see Figure 2D,E).

## 4. Discussion

The present study is the largest and most comprehensive to date phenomenological investigation of the human sense of boundaries (SB) and its dissolution. The results validate our previous studies [25,26,27], providing robust evidence that adept meditators can profoundly change the self-world structure of their experience. The prospects of trained participants volitionally altering their SB within lab settings, in conjunction with brain recordings, presents the field with a viable neurophenomenological paradigm for operationalizing processes giving rise to pre-reflective selfhood (i.e., self-specific processing, cf. [58]), and more generally, for investigating the fundamental structure of self-consciousness.

That the meditators succeeded in producing and holding such deep meditative states was evidenced by their epoch-based self-ratings of quality and stability and elaborated extensively in their phenomenological descriptions. The reported meditative experiences were diverse and ranged from common meditative concentration states in which SB was clearly present to altered states of consciousness in which the aspects commonly understood to be constitutive of the sense of pre-reflective self-dissolved—including sense of location, agency, first-person perspective, and bodily form. Despite this diversity, a single dimension emerged capturing a large part of the variance in these phenomenological categories, establishing meditative SB dissolution as a relatively uniform process with varying levels of depth. Furthermore, the extent of dissolution in this dimension depended on the employed technique and was correlated with previous meditative practice.

In the following discussion, the phenomenological findings are first grounded within the context of contemplative research. Then, based on the results, we deepen the exploration of the dynamic experiential structure of the SB and discuss its implications regarding the sense of self. We focus on (1) the dissociation of self-boundaries and body-boundaries, (2) the role of attentional disengagement in SB dissolution, and (3) the affective foundation shaping the SB relating oneself with the surrounding. We then discuss how our findings support an enactive approach to the self and highlight the value of our analytic methodology in promoting a neurophenomenological dialogue.

### 4.1. Phenomenological Mapping of Meditative SB Alteration

The large and diverse sample of SB alteration experiences enabled a meticulous examination of the generic experiential structures involved in such phenomena. The analysis of the phenomenological descriptions yielded six experiential categories (see Table 1) characterizing the types of experiences involved in meditation-induced SB alteration, and a classification of facilitating meditative techniques (Table 2). The described categories specify distinct, but partially overlapping, experiential features, which allow systematically differentiating participants according to typical experiences. Importantly, most of the categories (self-location, first-person perspective, sense of agency, body sensations) are in line with the SB dissolution characterization presented in our previous study [25], and are closely related to previous conceptualizations of the embodied self [3,9,10]. However, the large sample and variability in expertise allowed specifying two novel SB-related experiential categories (attentional disposition, affective valence).

Our investigation adds to the evolving scientific effort aiming at a deeper understanding of meditation’s experiential features and their underlying cognitive mechanisms [36,55,67,68,69,70,71,72]. Crucially, the accumulation of concrete and detailed first-person descriptive accounts, which are lacking in contemplative research (but see [25,36,67,70,73,74,75]), refine our earlier findings and may inform previously proposed heuristic frameworks of meditation. For example, the influential framework by Dahl and colleagues [71] suggests distinct families of attentional, deconstructive, and constructive practices. On the one hand, many of the meditative experiences described in our study seem to belong to the family of attentional meditation, which utilize processes related to the regulation of attention. Phenomenologically, this can include the manipulation of object orientation, aperture of attention, meta-awareness, effort, and control of the dynamics of attentional engagement and disengagement [54]. On the other hand, the described meditative states also relate to the deconstructive family, and more specifically, to non-dual-oriented insight practices, which “are designed to elicit an experiential shift into a mode of experiencing in which the cognitive structures of self/other and subject/object are no longer the dominant mode of experience” [71] (p. 519). Although Dahl and colleagues note the similarity in some attentional and deconstructive types of meditation, they emphasize their separability based on distinct mechanisms and aims, suggesting self-inquiry and insight as central in deconstructive practice. In contrast, the diversity and temporal unfolding of the SB dissolution meditative experiences rather points towards an overlap of, and dynamic flow between, employed techniques and engaged processes. Specifically, our findings indicate that certain modes of attention (in particular a stable and disengaged mode of attention) are an inherent part of non-dual deconstructive practices that target pre-reflective self features and the SB (as further discussed in Section 4.3). In addition, we found that imagination-based meditation may also contribute to SB dissolution, while the proposed taxonomy by Dahl and colleagues suggests such practice belongs to a distinct family of constructive meditation. Thus, while categorical taxonomies are essential for heuristic and instructive purposes, the dynamic and interdependent flow of mental processes during meditative practice may be more accurately described by multidimensional phenomenological schemes as presented here (see [54]) or by proper dynamic (diachronic) accounts as enabled by the micro-phenomenological method [50,73].

Considered more broadly in the context of Buddhist mindfulness practice, our investigation of meditative forms of SB alteration can elucidate the role of mindful awareness, self-inquiry, and insight involved in deconstructing pre-reflective features of the sense of self. Initially (as done in the preparatory training), a meditative inquiry of the SB may begin by quieting the mind and bringing mindful awareness to this pre-reflective feature of our conscious self-experience, making it more explicitly experienced as an object of reflection. Such meta-awareness can illuminate components of the SB and evidence its dynamic and flexible nature in changing circumstances (as elaborated in the next sections). In other words, simply being mindful of the SB de-reifies its apparent solidity and potentially reveals its dependence on attention, affect, mobility, intersubjectivity, and other factors. From a Buddhist perspective, this might be considered fruitful in terms of undermining commonly held views regarding oneself as unitary and unchanging, and eliciting insight into the nature of impermanence, no-self, and suffering [76]. In this context, deep meditative SB dissolution resulting from intense or highly skillful practice can be especially useful in deconstructing the more persisting structures of self-experience, which include the fundamental duality of subject and object. Interestingly, our findings suggest curating such deep meditative states through attentional disengagement and a release of intentional effort, may also destabilize meta-awareness (e.g., drifts into mind wandering). Thus, the precariousness of such meditative practices warrants further investigation regarding the type and degree of training, as well as supporting conditions, necessary for the cultivation of wholesome self-dissolution experiences.

The phenomenological account of SB dissolution also adds to a growing literature describing similar altered states of consciousness, such as non-dual awareness [77] and minimal phenomenal awareness [78] (for a comparison between these terms, see [79]). Crucially, our investigation contributes phenomenological detail regarding the dynamics related to entering and maintaining meditative alterations of consciousness, often lacking in theoretical discussions. This analysis delineates these changes by describing graded transformations of several pivotal experiential features of the structure of pre-reflective selfhood. The involvement of attentional disengagement, as well as affective factors, in shaping such altered states are discussed in the following sections.

### 4.2. The Intricate Dynamics of Self-Boundaries and Body-Boundaries

The phenomenological examination compels us to re-evaluate our early conceptualizations and further highlight the complexity and flexibility of the human experience of boundaries. The frontloading of two meditative SB alteration conditions was designed to explore the pre-reflective sense of self in two contrasting poles: SB dissolution (SB−) for inducing global self-attenuation, and an active SB enhancement (SB−) condition for maintaining or accentuating the sense of boundaries, thus emphasizing the pre-reflective sense of self. However, this approach may have been overly simplistic by not considering the intricacy of the SB. Although the distinction between the *sense* of boundaries and the physical body boundaries was previously articulated [25], the present findings shed light on the dynamics involved in SB alteration, disentangling the sense of self-boundaries from body-boundaries and showing its relation to the sense of self.

While self and body naturally coincide when naively and grossly considering the sense of pre-reflective embodied self, meditative practice can challenge this intuitive identification by bringing to the foreground its variations [36]. Two main trends regarding the SB were characteristic of each meditative condition. In the SB+ condition, participants were asked to maintain a clear sense of boundaries which they understood as referring to their body boundaries, resulting in most cases in the practice of a body scan meditation. The described experiences were mostly characterized by attention oriented toward bodily sensations accompanied by a constriction of the sense of space to the area of the body (within bodily boundaries), while disregarding and even blocking any notion of the surrounding peripheral space. In this kind of experience, the boundaries of the body were accentuated and became more closed and defined in terms of the quality of sensations and location. However, in terms of self-boundaries, a different change was reported. The sense of self in this condition was for some participants less pronounced compared to ordinary daily conditions of worldly engagement. Instead, it was often identified with a sense of an observer, with its location narrowly contracted and centered in the head, in the chest, or alternatively, in the drifting location of attention. It appears that contrary to our initial presupposition, meditative SB+ condition did not accentuate the sense of the pre-reflective embodied self but rather influenced self-boundaries by distinguishing them from body-boundaries. In its stillness, the body was experienced more prominently as an object perceived *by* an observing self (Körper), rather than directly lived through *as* a self (Leib) [33].

In the contrasting condition of SB−, participants described many different kinds of experiences, which differently influenced their sense of body- and self-boundaries. While for most participants, the form and sensations involved in body-boundaries became indistinct, their sense of self changed in a variety of ways. For some participants, the sense of self was still prominent in being clearly identified as an active agent, controlling mental activity and located in the center of experience, sometimes identified with the general location of the body. Other participants identified the limits of their sense of self with the expanding boundaries of attention, neglecting the body altogether. Such identification corresponds with experienced changes in the sense of personal ownership, often understood as the appropriation of experiences, thoughts or actions as being ‘mine’ [80] (Note that the sense of ownership was initially investigated in the interviews but ultimately dismissed during the analysis process, due to insufficient subtlety and deficient reliability often related to use of Buddhist jargon.) Finally, a distinguished group of adept participants reported an even deeper sense of SB dissolution in which many aspects of the structure of awareness became indeterminate, including, most notably, the intentional boundaries between perceiver and perceived. In other words, the fundamental dual subject-object structure of awareness, manifesting as a very basic sense of self-boundary, dissolved, and transformed into a non-dual experience of unitive space.

These results emphasize not only the range of SB flexibility, but also its intricacy and diversity in its diverse manifestations linked not only to the interactivity of the body but perhaps more accurately to the changing domain of active interaction within a given experience (e.g., body sensations, space, imagination, or attention). A complete conceptualization of such an experiential structure has to account for the many possible SB forms and inter-related factors such as body image, body scheme, affordances, and identification. The current investigation is thus only another initial step in understanding and specifying the structure of the SB as viewed through the unique scope of its dissolution in deep meditative states.

### 4.3. Attentional Disengagement (Letting Go) Facilitates SB Dissolution

A comparison of the two meditative conditions of SB alteration highlights the influence of attentional dynamics and the sense of agency over the structure of self-experience. These two categories clearly differentiated between the two meditative conditions (Figure 1B). Meditative experiences in the SB+ condition were mostly characterized by an active dynamic control of focused attention engaged in scanning body sensations while disregarding any distant external stimuli. In contrast, experiences in the SB− condition were mostly characterized by a distinctively wider scope of attention, a far less dynamically active disposition and a decreased orientation towards objects [54]. This pattern was further emphasized in deeper experiences of SB dissolution, which were often described by participants as resulting from surrendering into an effortless sense of passivity characterized by an attenuation of all mental activity, and more subtly, the suspension of attentional dynamics. This complete attentional disengagement relates to meditation technique #G, involving a mental gesture of relaxation, letting go, and release, which was found to be the most effective in inducing SB dissolution (Figure 2C). It is also associated with subcategories Attention#4 & Agency#3, which were correlated with higher meditative expertise (Figure 2E).

Such radical changes in the sense of agency typified the most distinctive SB transformations and were strongly linked with profound changes in the structure of experience (as indicated by the network analysis; see Figure 2). This includes the change in first-person perspective towards non-dual awareness (NDA, 1PP#3), a state characterized by the absence of the dualistic subject-object structure of experience [79]. Attentional disengagement and suspension of (mental) agency were also linked with significant changes in the quality of self-location and felt space, which rather than feeling expansive, tended to lack distinctive features to varying degrees (Location#4). The role of attentional disengagement was also evidenced in descriptions regarding the significance of stable concentration and thought processes. Some participants reported that the state of SB dissolution was dependent on attention not being drawn to any spontaneous mental content, such as thoughts which appeared and disrupted the delicate quality of the meditative state (e.g., “Suddenly a wandering thought, a distinct feeling, which comes with form, which comes with the loss of this open space”, M6). These can be related to the increasingly formless nature of such experiences, related to a high level of dereification [54].

As attentional regulation is central in meditative practice [67], we suggest that other meditative techniques reported in this study potentially influenced the quality of the SB (although to a smaller degree) by manipulating attentional dynamics indirectly. Specifically, the use of imagination of spatial scenes or the intentional recollection of past events (Technique#3) were often associated with a widening of the attentional scope and possibly with some moderation of its active dynamics. The more subtle form of attentional disengagement in these techniques can be discerned through their dynamic unfolding of SB dissolution. These descriptions emphasize the active use of a certain technique (such as imagining wide open sky) as an initial volitional gesture, which helps trigger a more stable and passive state in which sense of agency decreases and boundaries dissolve. Similarly, it seems that experiences in which attention was turned outwards (Technique#2) were effective to the extent that the change in object orientation was also accompanied with a change in the scope and dynamics of controlling attention. For instance, a practitioner who volitionally directed attention to wander around different areas of the surrounding space (M20) experienced a milder SB dissolution compared to a practitioner who turned attention “more and more upwards, opening very wide” until a stable state of black space-like stillness was attained (M43). Based on all the evidence provided above, we suggest that by coming to complete rest, attention ceases to perform its organizational role in structuring awareness, thereby promoting dissolution of the SB and of experiential form in general [81]. 

Of note, in mechanistic terms, these qualitative insights seem to be parsimonious with recent discussions of meditation within the predictive processing and active inference frameworks [82,83,84], where the release of attentional focus would correspond to a leveling of (ingrained and biased) precision weights, resulting in a (partial) suspension of ordinary self-evidencing (i.e. suspending the enactment of generative models [85]).

### 4.4. The Relation of Affect and Sense of Boundaries

The results speak to the essential link between affectivity and self-boundaries, previously implicated for its relevance in social cognition (e.g., [86]) and elaborated more broadly as part of the fundamental pre-reflective emotional background that structures our sense of relatedness to the world and to other individuals [16,87,88]. The findings indicate that meditative states directed towards SB dissolution were mostly experienced as more positive than those directed towards SB maintenance. This is aligned with our previous study [26]. Furthermore, emotions (both positive and negative) were generally described as stronger in intensity during the SB− condition. These reports, which ranged from distress to bliss, are in line with a recent study [36], outlining the adverse effects which may result from meditation-induced changes to the sense of self.

The findings shed some light on the dynamic correspondence between SB and affect. We suggest that the flexibility and permeability of self-boundaries, ranging from open and wide to closed and constricted, can be understood in relation to an innate concern for distinction (involving protection and autonomy) and participation (involving connectedness and openness) [13]. The two contrasting meditative conditions are helpful in briefly elucidating that. For example, experiences in which SB was accentuated were often positively appraised as solid, clearly defined, stable, under control or safe—satisfying a concern for protection or predictability. On the other hand, when negatively appraised, these experiences were described as tense, physically inconvenient, claustrophobic or stifling—pointing towards an unmet need for openness or adaptability. In meditative experiences of SB dissolution, there was frequently a sense of opening of awareness, release of tension and letting go of control. These experiences were often appraised very positively by the participants, but sometimes experienced as distressing and intimidating, related to one’s willingness or ability to give up control and, figuratively, lower one’s guards. As described by one participant, “there’s something very delicate, very innocent there—vulnerable” (M32). One way of understanding the ambivalence in affective response is as resulting from an amplified tension between distinction and participation during SB dissolution, leading to potentially blissful unification which in turn may be also highly precarious and exposing.

Thus, experiences of SB dissolution accentuate the sensitivity inherent in engaging with the world and interacting with others. The above-mentioned discussion points towards the wholesome role of flexible boundary dynamics in balanced affective reactivity. While the capacity to dissolve one’s sense of boundaries seems trainable, as suggested by a strong correlation between degree of dissolution and meditation experience (cf. Figure 2), the trainability of the affective valence of the arising SB dissolution state was not indicated, and warrants further investigation. Such investigation, describing the conditions constitutive of desirable states of dissolution, could have important implications for teachers of mindfulness and possibly also clinicians. These considerations, sometimes addressed by mindfulness teachers [89,90], are becoming increasingly relevant due to the phenomenological similarities between some psychopathologies (depersonalization in particular) and SB dissolution states, as well as the rapidly increasing pool of Western experienced meditators who are able to produce such states, sometimes with negative consequences to their well-being [36,43]. On the other hand, with proper adaptation and contextualization, the findings raise the possibility of designing specially-tailored interventions for ‘flexing’ the SB, potentially benefitting clinical populations suffering from psychopathologies involving marked alterations in minimal and embodied self-processes (e.g., PTSD [35,91,92] or depersonalization disorder [93]).

Another direction we are currently pursuing is examining the link between SB dissolution-induced affective valence and the fear of non-being. We hypothesize that enhanced death denial (as operationalized in [94]), may be associated with negatively valenced affect during SB dissolution. Thus, it may be the case that SB-dissolution training should be complemented with certain constructive practices such as death acceptance meditations, as commonly done within traditional Buddhist frameworks [95].

### 4.5. An Enactive Approach to the Self

The enactive approach offers a distinctive view of cognition as organized around core ideas of autonomy, sense-making, emergence, embodiment and experience (see [96,97,98] for overviews). In the context of the study of the self, the enactive approach emphasizes the different dimensions of the self-organizing or individuating nature of embodied activity which relates the organism to its environment, extending from homeostatic self-regulation to sensorimotor coupling with the surroundings to intersubjective interaction [99]. These active self-generating (i.e., autopoietic) processes are said to engender the “co-emergence of inside and outside, of selfhood and correlative world or environment of otherness” ([100], pp. 48–49). The novelty in such an enactive approach to subjectivity and selfhood is far-reaching and worth emphasizing: the pre-reflective sense of self is not separate from the process of perceiving and acting. Rather than an exclusively mental and internal affair (e.g., a passively existing cognitive model or representation), the sense of self is said to implicitly arise during attention-demanding interactions with the environment in which it is embedded and inherently related. This echoes the conception of a lived world (in german, *Umwelt*) which denotes the environment as directly experienced by an organism, elaborated in the phenomenological tradition by Edmund Husserl [33] and Maurice Merleau-Ponty [61,101,102].

The current results are aligned with this enactive view of the self by emphasizing the central roles of active attentional engagement and affect in the interactional dynamics essential to the SB and the co-emergence of selfhood with otherness. The reduction in sensorimotor and mental activity during meditation resulted in distinct changes in the structure of self-experience and of the experienced environment. This was specifically exemplified in the contrast between meditative conditions, accentuating the influence of distinct patterns of attentional dynamics in shaping the SB (discussed in Section 4.3). Maintaining a sense of boundaries (SB+), associated with dynamic attention oriented towards the body, confined spatial experience and constricted the sense of self-location, while maintaining the general structure of self-experience (discussed in Section 4.2). This experiential change is aligned with the centrality of sensorimotor activity in the enactive view, but even more critically, with that of attention-demanding interaction which has been characterized as self-specifying [61]. Here attention is regarded as an active self-regulatory process that tunes the organism’s contact with the world (similar to how eye-movements would do) and thereby induces a difference between external (afferent) and self-induced (re-afferent) changes. The attentional disengagement, which was associated with deep SB dissolution experiences, provides further support for this suggestion. Such dissolution experiences are characterized by unique changes that not only radically alter the sense of self, but involve all experiential structures, including a dissolution of surrounding space. We suggest interpreting such results in light of the enactive approach, which predicts that such unusual reduction in the incessant activity of perception, action and, more subtly, attention will undercut the contribution of these sense-making mechanisms to the processes from which self and world co-emerge. Additionally, echoing a social enactive perspective [88], we have highlighted in the previous section the inherent affective tension essential to shaping self-boundaries, which define oneself as an autonomous yet interrelated being.

### 4.6. Operationalizing Phenomenology within the Neurophenomenological Context

The systematic organization of the phenomenological results renders it a powerful tool for guiding subsequent neural analyses, thus creating a fruitful neurophenomenological dialogue. As demonstrated by Lutz ([29]), one way this can be done is by creating a phenomenological mapping, which captures the existing experiential variability, in turn bringing forth and thus accounting for variability in neuronal responses otherwise ‘averaged out’ as noise. Importantly, the results indicated that most of the variation in the reported phenomenology was explained by the single dimension of SB dissolution, accounting for 58% of the variance in the entire phenomenological space, with strong loadings on all dimensions except affect. Furthermore, a summary score based on the intercorrelated dimensions (DD_C_) correlated nearly perfectly with a holistic dissolution rating (DD_R_), indicating that a single dimension encompassing all phenomenological dimensions (except valence) captures meditative SB dissolution comprehensively. While, post factum, these findings might not seem surprising, different clusterings and structurings of the phenomenological space would have been plausible, such as different groups of agentic, perspectival or bodily dissolvers. While theoretical implications of this finding are discussed below, it also has more practical implications in terms of constraints to subsequent neural analyses. Mainly, it suggests focusing on a group of participants experiencing deep global dissolution of their sense of self, as indicated by high scores on all five dissolution dimensions (passive agency, indeterminate location, non-dual FPP, formless attention, imperceptible body). Conversely, in the “with boundary” condition (SB+), a proto-typical group emerged, characterized by default values on all dimensions (active agency, location within body, normal FPP, focused attention, prominent body). Accordingly, both groups have been specified in our pre-registered analyses (see https://osf.io/bsxua/registrations; accessed on 17 November 2020). Additionally, some (partial) independence of the Location and Agency dimensions was observed in the SB- condition (cf. Figure 2B), suggesting potentially different “routes” to boundary dissolution, characterized broadly by active space-driven SB dissolution and, alternately, passive formless SB dissolution. Disentangling these two routes on the neural levels can be done by contrasting these dimensions, or by creating specifically defined clusters reflecting such contrast. Similarly, the meditative techniques of “turning attention outwards” as well as “relaxation, release, passivity” emerged as relatively frequent (cf. Figure 1C) and effective (cf. Figure 2C) and could be contrasted neurally to disentangle phenomenologically dissimilar participants within specific aspects, thus refining the neural analysis and exploring potentially distinct neuro-cognitive mechanisms. Finally, the independence of the affective dimension may indicate neural processes correspondingly independent from those underlying SB dissolution.

The presented quantitative work exemplifies how phenomenological descriptions can be mathematized to formally describe relationships between different dimensions, as well as with other external variables. Such mathematization is an important step in the neurophenomenological dialogue, and more generally in naturalizing phenomenology [7,103]. Note, however, that different varieties and arguments about the limitations of naturalizing phenomenology exist [104,105,106]. In the present case, the computed score for degree of dissolution (DD_C_) illustrates both the advantages and limitations of this process. On the one hand, it yields a transparent and explicit calculated SB dissolution score. On the other hand, it lacks the intuitive face validity of the holistic rating, as well as, potentially, sensitivity to subtle cues and gestures not captured in the extracted categories. However, the tight convergence of both scales in the present case supports the formalization of SB dissolution as a combination of features captured in the five categories.

Opening up another bridge for the neurophenomenological dialogue, it has recently been emphasized that (en-)active inference and predictive processing offer a formal (computational) model for understanding the embodied and enactive mind [85,107]. The experiential reports portrayed here indicate that central tenets of this model, including self-evidencing and precision weighting, are deeply modulated in these states, thus opening up an interesting approach to specifying phenomenological and neuronal counterparts of these formulations.

In sum, the operational design of the phenomenological investigation demonstrates the development of empirical first-person methodologies in the context of a neurophenomenological research paradigm, a dialogue to be continued in future neurophysiological work.

## 5. Conclusions

This study presents the largest to date phenomenological mapping of the sense of boundaries (SB), a central feature of conscious experience highly relevant for the study of (self-)consciousness in general, and the study of meditation in particular. We replicated previous results establishing that such subtle yet profound altered states of self-experience can be produced and reproduced under robust experimental settings. We also outlined, for the first time in the literature, the effect of meditation technique and attentional dynamics on SB phenomenology, as well as the latter’s complex relation with affective valence. While strong associations with lifetime meditative practice are suggestive, the results call for future longitudinal studies investigating causality of practice and moderating factors (e.g., intensity and type of practice), as well as mediating effects on adverse experiences, wellbeing, insight, and social connection. Finally, we illustrate by example the process by which deep phenomenological information can be collected, processed, and formalized in such a way as to allow quantitative analyses and inform neural analyses, which will be implemented in our future work. Overall, the present study highlights the unique potential of neurophenomenology to advance the scientific study of consciousness and self.

## Data Availability

The processed phenomenological data (categorical characterizations) presented in this study are openly available on the Open Science Framework (OSF) project page at https://doi.org/10.17605/OSF.IO/BSXUA (accessed on 1 April 2021). The raw interview transcripts in Hebrew language are available on request from the corresponding author.

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
