# Peer review of "Self-Boundary Dissolution in Meditation: A Phenomenological Investigation"

_brainsci, 2021, doi:10.3390/brainsci11060819_

Round 1
Reviewer 1 Report
Authors in this paper conducted a comprehensive phenomenological inquiry into meditative self-boundary alteration. The induced states were systematically characterized by changes in six experiential features including the sense of location, agency, first-person perspective, attention, body sensations and affective valence, in addition with employed meditative techniques and overall degrees of dissolution.
My comments to the article are as follows:
- As part of the Introduction, I propose to expand the background by referring to the verification of states of mind. For example, you can cite the item: Project and simulation of a portable device for measuring bioelectrical signals from the brain for states consciousness verification with visualization on LEDs, Advances in Intelligent Systems and Computing, Springer. I think that it would also be worthwhile to extend the background of the article with a fundamental reference to the sources of potentials in the human brain, including, for example, the following: Characteristics of Question of Blind Source Separation Using Moore-Penrose Pseudoinversion for Reconstruction of EEG Signal, Automation 2017: Innovations In Automation, Robotics And Measurement Techniques, Book Series: Advances in Intelligent Systems and Computing, Springer.
- Please explain on what basis the selected group of study participants.
- I propose to include the largest tables (No. 3 to No. 11) as attachments to the article, because they take up a lot of space in the text, which makes the article itself very long.
- As part of the Conlusions, I propose to include plans for the future.
- The graphs / characteristics presented in the article, for example in Fig. 1 and Fig. 2, should have axis descriptions based on units of measurement. Please complete this.
- I suggest that you do not use the Bottom Props on page 5.
Reviewer 2 Report
The article present an interesting study with a novel phenomenological approach linked to a dedicated meditation programme, to investigate the effects of meditative practice on the pre-reflective self, in terms of self boundary (SB) dissolution. A comprehensive and thorough phenomenological inquiry is conducted in the study. Six experiential features (categories) linked to the pre-reflective self were investigated. A unitary dimension of boundary dissolution, linked to the meditative process of "letting go", emerged in the relationships between the experiential categories. An interesting enactive approach to the self is thus proposed.
The proposed approach appears relevant for both meditation science and an increased understanding of the self, with potential clinical implications.
The article is very well written, reporting a study with accurate methodology and data analysis, with a clear report of the results, and with useful supplementary material describing in detail the implemented mental / meditation training.
The results are interesting and novel, consistent with previous work from the same group, and are discussed with depth and focus.
The following aspects can be further considered to contextualize properly and potentially increase the impact of the study:
- Discuss the implications of the proposed emphasis on attentional aspects in respect to the influential taxonomy of meditation practices by Dahl et al. (2015). This is considered in the Discussion, but seemingly not in depth. In particular, how attentional meditation practices can be interrelated with self deconstructive practices, given that Dahl et al. suggest distinct categories of practices for attentional meditation practices and self deconstructive practices, whereas the present study highlight particular attentional practices as self deconstructive, even at the level of dissolving pre-reflective self features/categories. These implications can be highly relevant in the field of meditation / contemplative science.
2. Discuss in more depth the implications of the study for the pre-reflective self concept, and how the reflective aspect related to mindfulness meditation practice relates to the pre-reflective self and the self boundary features. What is the difference between a pre-reflective self experience without meta-awareness and a mindful (non-narrative) self experience, including about self boundaries?
3. How can the findings in the study relate to adverse effects of meditation practices?
4. Could a phenomenological matrix of pre-reflective self experiences in meditation be constructed, e.g. by six dimensions corresponding to the six investigated features of pre-reflective self experience, and three other dimensions corresponding to emotional valence, mindfulness and equanimity? Indeed it can be hypothesized that loss of self boundaries is wholesome only in presence of mindfulness and equanimity, and probably other mental factors (e.g. the so called factors of enlightenment, which can also be understood as factors supporting "letting go" with mental/emotional balance), with a higher likelihood of adverse effects of meditation without the support of such wholesome balancing mental factors.
Please also discuss how the fundamental notion of "insight" in meditation (a core aspect of self deconstructive practices) relates to the notion of SB dissolution.
5. Discuss the implications of the proposed approach for longitudinal studies, e.g. in intensive meditation retreats.
Round 2
Reviewer 1 Report
Dear Authors,
Thank you very much for your comprehensive answers to my questions.
The article has been improved to the extent suggested by me.